# Fine-Grained Building Classification in Rural Areas Based on GF-7 Data

**DOI:** 10.3390/s25020392

**Published:** 2025-01-10

**Authors:** Mingbo Liu, Ping Wang, Peng Han, Longfei Liu, Baotian Li

**Affiliations:** 1National Disaster Reduction Center of China, Ministry of Emergency Management of the People’s Republic of China, Beijing 100124, China; wangping@ndrcc.org.cn (P.W.); hanpeng@ndrcc.org.cn (P.H.); liulongfei@ndrcc.org.cn (L.L.); 2Bureau of Emergency Management of Pingquan City, Pingquan 067500, China; pqyingji@163.com

**Keywords:** GF-7 data, building classification, building height, rural areas

## Abstract

Building type information is widely used in various fields, such as disaster management, urbanization studies, and population modelling. Few studies have been conducted on fine-grained building classification in rural areas using China’s Gaofen-7 (GF-7) high-resolution stereo mapping satellite data. In this study, we employed a two-stage method combining supervised classification and unsupervised clustering to classify buildings in the rural area of Pingquan, northern China, based on building footprints, building heights, and multispectral information extracted from GF-7 data. In the supervised classification stage, we compared different classification models, including Extreme Gradient Boosting (XGBoost) and Random Forest classifiers. The best-performing XGBoost model achieved an overall roof type classification accuracy of 88.89%. Additionally, we proposed a template-based building height correction method for pitched roof buildings, which combined geometric features of the building footprint, street view photos, and height information extracted from the GF-7 stereo image. This method reduced the RMSE of the pitched roof building heights from 2.28 m to 1.20 m. In the cluster analysis stage, buildings with different roof types were further classified in the color and shape feature spaces and combined with the building height information to produce fine-grained building type codes. The results of the roof type classification and fine-grained building classification reveal the physical and geometric characteristics of buildings and the spatial distribution of different building types in the study area. The building classification method proposed in this study has broad application prospects for disaster management in rural areas.

## 1. Introduction

Fine-grained building type information is widely used in risk and loss assessment of natural disasters, such as earthquakes, mudslides, and floods [1,2,3,4,5]. It is also a key factor in studies on urbanization, energy consumption, and population modelling [6,7,8].

Building types can be obtained through field surveys, which are labor- and time-intensive. However, advances in remote sensing technology have made it possible to quickly obtain information about building types over large areas [9]. High-resolution multispectral remote sensing images can be used to extract color and shape features of buildings [10], while Light Detection and Ranging (LiDAR) provides information on building heights [11].

Some researchers have performed building classification by integrating high-resolution multispectral imagery with LiDAR. For example, Huang et al. (2017) classified buildings in Guangzhou into villa, apartment, low-rise, and non-building types based on building height information from LiDAR and textural, spectral, and geometric information from high-resolution remote sensing imagery, achieving an accuracy of 98.15% in the study area and 87.50% in the validation area [12]. Similarly, Ghasemian et al. (2024) used Google Street View (GSV), LiDAR-derived features, and high-resolution orthophotos to classify buildings in Vancouver and Fort Worth into apartment, house, industrial, institutional, mixed residential/commercial, office building, retail, and other classes, with an overall accuracy of 75% [13]. These studies demonstrate the feasibility of combining high-resolution multispectral data with building height data for building classification. Other studies have used open-access geospatial vector data, such as OpenStreetMap (OSM), Google Maps, and Baidu Maps, as well as human activity data, such as nighttime light data and geo-referenced social media data, to classify buildings. For example, Atwal et al. (2022) classified buildings into residential and non-residential classes in suburban and urban areas based on geometric features and associated tags extracted from the OSM dataset, achieving an overall accuracy of 96.73% to 98.02% [14]. Abhilash et al. (2022) classified buildings in Germany based on features from OSM and other auxiliary data, achieving a percentage prediction error of 3.64% for residential buildings, 13.14% for single-family houses, and −15.38% for multi-family houses [15]. Häberle et al. (2022) fused geo-referenced tweets and high-resolution remote sensing imagery to classify buildings in San Francisco, New York City, Los Angeles, and Washington, D.C. into commercial and residential classes, achieving an overall accuracy of 64% to 78%. Their results show that fusion frameworks can improve the classification of buildings [16]. In building classification studies, supervised machine learning methods, such as XGBoost and Random Forest, deep learning methods, and unsupervised clustering methods are often used to build models [17,18,19].

However, despite many achievements, there are still limitations to existing research, particularly in disaster management scenarios. One of the challenges is the timeliness, quality, and spatio-temporal consistency of data from different sources. For example, disaster loss assessment requires building type products that are based on the latest data and cover specific areas. This is challenging for methods that integrate high-resolution multispectral imagery and LiDAR data, as LiDAR data is often acquired by airborne platforms with limited coverage and is difficult to update frequently. Similarly, widely used geospatial vector data, such as OSM, suffer from incomplete attributes and significant variations in data quality [20]. Another issue often overlooked is that most studies have been conducted in urban areas, with few focusing on rural areas. This is partly due to the scarcity and obsolescence of high-resolution building footprint and height data in rural areas. Fine-grained building classification in rural areas is urgently needed because natural disasters frequently occur in rural areas, and information on building types can help assess vulnerability and loss.

To address these limitations, this study aims to develop a method for fine-grained classification of buildings in rural areas using regularly accessible multi-dimensional data from a single spaceborne platform. We utilized China’s GF-7 civil stereo mapping satellite to simultaneously acquire high-resolution multispectral imagery and building height information [21]. A hybrid method combining supervised classification and unsupervised clustering was developed to build the model based on the characteristics of the buildings in the study area.

## 2. Study Area and Data

### 2.1. Study Area

The study area is located in the northern part of Pingquan, Hebei Province, China (118°32′43″ E~118°52′20″ E, 41°2′27″ N~41°17′49″ N), with a continental monsoon climate (Figure 1a). It has an area of 562 km^2^ and altitudes ranging from 520 m to 1181 m above sea level. The main land cover types are forest, shrub, farmland, buildings, and water bodies. The main building types are bungalows, greenhouses, workshops, and multi-story buildings. The southern part of the study area is close to the outskirts of Pingquan City and is densely populated with many factories and residential areas. Another densely populated area is a town center in the northeast, which is a river confluence area with a relatively flat and open terrain and a large number of greenhouses. To the southwest, there is a large mine surrounded by tailings ponds and industrial facilities. The rest of the study area is mostly mountainous, with complex terrain and buildings located in various valleys. The diversity of topography and industry makes the study area a representative sample of rural areas.

### 2.2. GF-7 MUX Image

The GF-7 satellite was launched in November 2019 and is primarily used for natural resource monitoring and land surveying. The satellite operates in a 500 km sun-synchronous orbit, with a temporal resolution of 59 days and a ground swath wider than 20 km. It is equipped with a 26° forward-looking (FWD) and −5° backward-looking (BWD) dual-line array scanner and a backward-looking multispectral (MUX) camera. The FWD and BWD panchromatic images have a spectral range of 0.45 to 0.9 μm, with spatial resolutions of 0.8 m and 0.65 m, respectively. The MUX multispectral image has a spatial resolution of 2.6 m, and the spectral ranges of its four bands are 0.45 to 0.52 μm, 0.52 to 0.59 μm, 0.63 to 0.69 μm, and 0.77 to 0.89 μm, respectively [22,23]. The GF-7 data used in this study were acquired on 3 April 2020 and downloaded from the China Center for Resources Satellite Data and Application (CRESDA; https://data.cresda.cn/#/home, accessed on 22 October 2024).

The DSM extracted from the GF-7 FWD and BWD stereo images was used to perform the geometric correction of the MUX image [21]. The radiometric calibration and spectral response parameters provided by CRESDA were applied for radiometric correction. Radiance was converted to surface reflectance using the FLAASH atmospheric correction. Pansharpening was performed with the Gram-Schmidt algorithm to enhance spatial detail. The pre-processed MUX multispectral image (Figure 1b) was then used for subsequent classification.

### 2.3. GF-7 Building Footprint

Building footprints were extracted using the GF-7 MUX multispectral image and Mask R-CNN instance segmentation. Extensive manual correction of the extracted building footprints was performed through visual interpretation of the multispectral image [21,24,25,26]. A total of 50,925 building footprints were obtained in the study area, with a mean size of 154.52 m^2^, a median size of 75.75 m^2^, and a total area of approximately 7.87 km^2^. The building footprint was used as the basic unit for building classification and for calculating shape indicators.

### 2.4. GF-7 Building Height

The GF-7 building heights were extracted from the GF-7 stereo image. We proposed a building height extraction method that combines photogrammetry and deep learning (DELaMa) [21]. First, the difference between the GF-7 DSM and the DTM produced by DELaMa was calculated to obtain the normalized digital surface model (nDSM). The 90th percentile value of the nDSM within the building footprint was then used as the building height. The 90th percentile is an empirical parameter chosen to minimize interference from accessory structures on roofs, such as solar equipment, chimneys, and decorative structures, while approximating the roof’s highest point. The extracted building heights were highly consistent with reference building heights measured from the ICESat-2 LiDAR point cloud, with an R^2^ of 0.83, an MAE of 1.81 m, and an RMSE of 2.13 m. The GF-7 building heights were used for the fine-grained building classification.

## 3. Methodology

### 3.1. Overview

In this study, we used a two-stage method combining supervised classification and unsupervised clustering for fine-grained building classification [19,27]. The classification workflow is shown in Figure 2. First, we classified the buildings into different roof types using the supervised Extreme Gradient Boosting (XGBoost) classification model. Cluster analysis was then performed on buildings with different roof types based on their color and shape characteristics. Statistical indicators and expert knowledge were used to determine the number of clusters. Simultaneously, the heights of pitched roof buildings were corrected. We proposed a template-based height correction method that combined the building height information derived from the GF-7 stereo image with the slope angle of the pitched roof and the shape parameters of the building footprint. Finally, the buildings in the study area were coded into fine-grained categories based on the results of the roof classification, color and shape clustering analyses, and building height information. The experiment was conducted on a computer equipped with an NVIDIA^®^ GeForce RTX 3090 Ti GPU and 64 GB of memory.

### 3.2. Supervised Roof Classification

The Extreme Gradient Boosting (XGBoost) classification model creates a series of sequential decision trees. Each subsequent decision tree is constructed to minimize the bias of the previous tree, allowing the model to combine several weak learners into a strong classifier. XGBoost incorporates regularization and early stopping to prevent overfitting of the training dataset. Compared to Random Forest, XGBoost offers greater control over hyperparameters and is more complex. The XGBoost classification model is highly scalable and efficient, with excellent prediction performance. It has been widely recognized in numerous machine learning and data mining challenges, as well as in various studies and applications, including building classification [28,29]. We manually labeled 1171 samples of different roof types, including pitched, greenhouse, color steel, flat, and complex roofs. The median reflectance of the panchromatic and multispectral bands, the standard deviation of the panchromatic reflectance within each building footprint, and the area, length, width, and aspect ratio of the minimum bounding rectangle of the building footprint were used as explanatory variables. The median value was chosen to reduce the effects of shadows and building footprint distortions. Twenty percent of the labeled samples were used for validation to evaluate the model’s performance. In this study, we also evaluated the performance of the Random Forest model [30]. The Random Forest model is widely used in various classification tasks. It creates numerous independent decision trees, each based on a random subset of the training data and variables. Each tree makes its own predictions. The model takes into account the votes from all the decision trees to predict or classify unknown samples. The final prediction is based on the entire forest, helping to avoid overfitting.

Grid search was used to compare the effects of different parameters. The tuning parameters for the XGBoost model include the number of trees, the maximum tree depth, the number of randomly sampled variables, the learning rate, and the L2 regularization. The tuning parameters for the Random Forest model include the number of trees, the maximum tree depth, and the number of randomly sampled variables.

### 3.3. Evaluation Indicators

The importance of each explanatory variable in the XGBoost model was assessed using the variable importance indicator. This indicator represents the relative contribution of an explanatory variable to the model and is calculated by summing the gain of all the splits where the explanatory variable is used. The performance of the classification model was evaluated using the confusion matrix and statistical indicators such as *Recall*, *Precision*, and *F*1*-Score* as follows:(1)$$Recall=\frac{TP}{TP+FN}$$(2)$$Precision=\frac{TP}{TP+FP}$$(3)$$F1\text{-}Score=\frac{2\times TP}{2\times TP+FP+FN}$$
where *TP*, *FP*, and *FN* represent the true positive, false positive, and false negative predictions for a given category, respectively. The *F*1*-Score* is the harmonic average of *Precision* and *Recall*, which can better reflect the performance of the model on unbalanced datasets and is widely used in building classification research. We optimized the parameters of the XGBoost and Random Forest models using *F*1*-Score* metrics as evaluation criteria.

The McNemar’s test [31] was used to determine if there is a significant difference in the performance of the XGBoost and Random Forest models as follows:(4)$$Z=\frac{f_{XR}-f_{RX}}{\sqrt{f_{XR}+f_{RX}}}$$
where $f_{XR}$ is the number of samples correctly classified by the XGBoost model and misclassified by the Random Forest model, and $f_{RX}$ is the number of samples correctly classified by the Random Forest model and misclassified by the XGBoost model. At a significance level of α = 0.05, −1.96 < *Z* < 1.96 indicates that there is no significant difference between the two models, while *Z* > 1.96 indicates that the XGBoost model outperforms the Random Forest model and *Z* < −1.96 indicates the opposite.

### 3.4. Template-Based Height Correction

In the process of extracting building heights using the GF-7 stereo image, due to the limited resolution and lack of texture, stereo matching failed or had very low confidence at some locations, and the elevation values at these locations could not be used to extract building heights. For flat roof buildings, the effect of invalid data is not significant because we used the 90th percentile of the nDSM within the building footprint as the building height. However, for pitched roof buildings, particularly small bungalows, invalid data introduces a significant bias because the remaining valid data may be distributed across different parts of the roof and correspond to different heights (Figure 3b). Additionally, pixels on the roof are susceptible to interference from varying lighting conditions and lack of contrast, resulting in incorrect stereo matches and elevation values. To solve these problems, we developed a template-based height correction method for pitched roof buildings. In this method, the corrected height $H_{C}$ of a pitched roof building consists of two parts: the roof height $H_{R}$ and the eave height $H_{E}$ (Figure 3d). The roof height $H_{R}$ is calculated from the slope angle θ and the width of the minimum bounding rectangle. The eave height $H_{E}$ is the 90th percentile of the nDSM within the eave zone as follows:(5)$$H_{C}=H_{R}\times 0.9+H_{E}$$(6)$$H_{R}=\frac{Width}{2}\times \tan\theta$$(7)$$H_{E}={nDSM}_{90th}\;in\;eavezone$$
where the eave zone is defined as the area within the minimum bounding rectangle where the distance from the centerline is greater than 1/3 and less than 1/2 of the width (Figure 3b). An $H_{E}$ of less than 1.5 m is considered an outlier and is replaced by 1.5. The slope angle θ is determined from photos of the buildings in the study area. We collected street view photos of pitched roof buildings throughout the study area and estimated the roof slope angles (Figure 3c). The average slope angle was used as the roof slope angle in the template. Since the heights of the other buildings in the study area were defined as the 90th percentile of the nDSM within the footprint, and since we used the 90th percentile of the ICESat-2 LiDAR point cloud heights to validate the GF-7 building heights, the $H_{R}$ was multiplied by a factor of 0.9 to maintain consistency.

### 3.5. Unsupervised Cluster Analysis

The k-means clustering algorithm was used to distinguish different buildings with the same roof type [32]. The cluster analysis was performed in color and shape dimensions, respectively. Color and shape features have practical significance in application scenarios. For example, in our study area, low and dark roofed bungalows are mostly old and abandoned dwellings, while long strip bungalows are more likely to be used for industrial and commercial rather than residential purposes. Variables in the color dimension include the four bands of the multispectral image (B1 to B4) and the standard deviation of the panchromatic band in the building footprint (PanStd). The shape dimension variables include the area, length, width, and aspect ratio of the minimum bounding rectangle. The values of the variables were standardized. The $R_{variable}^{2}$ index was calculated for each variable to assess its significance in the cluster analysis as below:(8)$$R_{variable}^{2}=\frac{TSS-ESS}{TSS}$$
where TSS is the sum of squares of deviations from the global mean value, and ESS is the sum of squares of deviations from the mean value for each cluster. The number of clusters was determined by calculating the Calinski-Harabasz index (CHI) for 2 to 30 clusters [33]. The largest Calinski-Harabasz index values indicate solutions that perform best at maximizing both within-cluster similarities and between-cluster differences as below:(9)$$CHI=\frac{R_{CHI}^{2}}{n_{c}-1}\times \frac{n-n_{c}}{1-R_{CHI}^{2}}$$(10)$$R_{CHI}^{2}=\frac{SST-SSE}{SST}$$(11)$$SST=\sum_{i=1}^{n_{c}}\sum_{j=1}^{n_{i}}\sum_{k=1}^{n_{v}}{\left(V_{ij}^{k}-\bar{V^{k}}\right)}^{2}$$(12)$$SSE=\sum_{i=1}^{n_{c}}\sum_{j=1}^{n_{i}}\sum_{k=1}^{n_{v}}{\left(V_{ij}^{k}-\bar{V_{i}^{k}}\right)}^{2}$$
where n is the number of samples, $n_{c}$ is the number of clusters, $n_{i}$ is the number of samples in cluster i, $n_{v}$ is the number of variables, $V_{ij}^{k}$ is the value of the $k_{th}$ variable of the $j_{th}$ sample in the $i_{th}$ cluster, $\bar{V^{k}}$ is the mean value of the $k_{th}$ variable, $\bar{V_{i}^{k}}$ is the mean value of the $k_{th}$ variable in cluster i. If the Calinski-Harabasz index continues to increase with the number of clusters, or indicates too many categories, the number of clusters is manually specified.

### 3.6. Coding of Fine-Grained Building Types

The roof, color, shape, and height features were represented by the letters R, C, S, and H. R1 to R5 represent pitched, greenhouse, color steel, flat, and complex roof types. Building heights were classified as H1 to H4, representing less than 6 m (1 storey), greater than or equal to 6 m and less than 9 m (2 storeys), greater than or equal to 9 m and less than 12 m (3 storeys), and greater than or equal to 12 m (4 storeys and above). Fine-grained classification results were represented by the combination of roof, color, shape, and height types. For example, R1C2S3H1 means that the building has a pitched roof, and in pitched roof buildings, it belongs to cluster 2 in the color dimension and cluster 3 in the shape dimension, and its height is less than 6 m.

## 4. Results

### 4.1. Roof Types in the Study Area

Quality checks and corrections were made to the roof classification results. Roofs that could not be clearly identified in the image were marked as unclassified. Buildings with a footprint area of less than two-thirds of the minimum bounding rectangle were also considered irregularly shaped and marked as unclassified. Samples of different roof types are shown in Figure 4. Pitched roofs have a distinct ridge line and are usually orange, although some older pitched roof buildings are dark grey. Greenhouses have elongated shapes and curved surfaces and are often aggregated in patches. Color steel roofs are typically blue, red, or grey, with minimal slope. Some color steel buildings have large footprints. Flat roofs have little texture and relatively high reflectance. Complex roofs have multiple ridge lines and are mostly multi-storey.

The results of the roof classification after quality control are shown in Figure 5. The most common roof types are pitched roofs and flat roofs, accounting for over 60% of the total, followed by greenhouses and color steel roofs. Buildings with complex roofs are the least common, accounting for only 0.43% of all buildings. Pitched roof buildings are mostly dwellings, distributed throughout the study area and clustered in patches. Flat roof buildings are mainly outbuildings, which are numerous but more spatially dispersed. Some of the larger flat roof buildings are multi-storey office buildings belonging to schools, hospitals, government, and commercial organizations, mainly in the town center in the northeast and in the south near the outskirts of Pingquan City. The greenhouses are located on flat farmland and are clustered in patches. Some color steel roof buildings are small outbuildings mixed with pitched roof residential buildings, while others are workshops away from residential areas and large facilities around the mine. There are many workshops in the town center in the southernmost part of the study area. Complex roof apartments are also located in town centers.

### 4.2. Height of Buildings with Different Roof Types

Based on the results of the roof type classification, we checked the accuracy of the building height for different roof types (Figure 6a and Figure A1). The number of buildings with the five different roof types in the validation set is 159, 90, 41, 23, and 16, respectively. ICESat-2 LiDAR point cloud data were used to validate the heights of the buildings. We checked the spatial distribution of the ICESat-2 LiDAR beams on the GF-7 high-resolution image and the vertical profile of the point cloud to ensure the quality of the validation [21,34,35,36]. Pitched roof buildings and greenhouses have the worst height accuracy. The validation results for color steel, flat, and complex roof buildings show MAEs of no more than 1.5 m and RMSEs of less than 2 m. The failure of height extraction for greenhouses was mainly due to their curved surface and lack of texture. However, the height of the greenhouses in the study area is relatively homogeneous. According to the ICESat-2 LiDAR point cloud measurements, the mean height of the greenhouses is 3.84 m with a standard deviation of 0.84 m. Therefore, we set the height of all 5223 greenhouses to 3.84 m. Although the pitched roof building heights have an RMSE of more than 2 m, unlike the greenhouses, they show a correlation with the ICESat-2 reference building heights. We collected 15 street view photos of pitched roof buildings in different zones of the study area to estimate the roof slope angle for our template-based height correction. The average slope angle was calculated as 32.5°. After correction, the MAE and RMSE were reduced to 0.91 m and 1.20 m, respectively, the lowest of all roof types (Figure 6b). The R^2^ also improved from 0.56 to 0.58. The heights of all 16,452 pitched roof buildings were corrected.

### 4.3. Results of the Fine-Grained Building Classification

A total of 41,160 buildings were classified into 140 categories based on a combination of roof types, colors, shapes, and heights (Figure 7). The number of buildings in these categories is highly unbalanced. 77 categories have fewer than 10 buildings, 113 categories have fewer than 100 buildings, and the 10 most common categories contain more than 75% of the buildings. The images of the 10 most common building types are shown in Figure 8, together with images of 10 other representative building types. The buildings were placed in the center of the images and the cartographic scale is the same for all images to facilitate comparison. The most common building type in the study area is R1C1S3H1, i.e., small single-storey buildings with light-colored pitched roofs, accounting for 18.17% of all classified buildings. This type of building is the residence of local villagers and can be found throughout the study area. R1C1S3H1 and R1C1S1H1, the medium sized single-storey buildings with light-colored pitched roofs, are the dominant building types in remote valleys. R4C2S1H1 and R4C2S2H1 are the most common small outbuildings in residential areas, accounting for 20% of all classified buildings. R2C2S2H1, the small dark-colored greenhouse, is the most common greenhouse type and the third most common building in the study area. The R1C1S1H2 and R1C1S1H1 types are similar in appearance and contain a similar number of buildings, but the R1C1S1H2 buildings are taller. The R1C1S1H2 is the most common building type with a height of 6 m or more. R1C2S3H1, the small single-storey buildings with dark pitched roofs, account for 5.39% of all classified buildings, most of which are old houses. Most color steel roof buildings are small blue-colored outbuildings of type R3C1S1H1. Multi-storey buildings, such as R3C1S4H4, R4C3S3H4, and R5C1S2H4, are likely to be factory, office, and apartment buildings. These types of buildings are few in number and are mainly found in town centers in the northeastern and southern parts of the study area.

## 5. Discussion

### 5.1. Comparison of Different Supervised Classification Models

The confusion matrices of the XGBoost and Random Forest classification results are shown in Figure 9, where the horizontal axis represents the categories predicted by the model, the vertical axis represents the actual categories, and the values on the diagonal indicate the number of samples the model predicted correctly. From the confusion matrices, the accuracy for all categories is 88.89% and 84.19% for the XGBoost and Random Forest models, respectively. The *Recall*, *Precision*, and *F*1*-Score* of different classification models are shown in Table 1. The XGBoost model outperformed the Random Forest model on all evaluation metrics except for the *Precision* of the pitched roof. Both models gave the highest *F*1*-Scores* for the pitched roof type, followed by the greenhouse type, and the lowest *F*1*-Scores* for the color steel type. A proportion of the color steel roof buildings were classified as flat roof buildings in both models, suggesting that some of the color steel roofs have spectral and shape characteristics very similar to flat roofs. The *Z* value of the XGBoost and Random Forest models for pitched, greenhouse, color steel, flat and complex types are 1.73, 1.41, 0.63, 0.82 and 1.41 respectively, indicating that there is no significant difference between the performances of these two models in classifying specific roof types. However, the overall *Z* value for all types is 2.29, indicating that the overall performance of the XGBoost model is better than the Random Forest model.

The variable importance index suggests that PanStd, area, and width are the three most important variables in the XGBoost classification model, followed by band 1, band 3, and panchromatic reflectance, aspect ratio, length, and band 4 reflectance. The optimal parameters for XGBoost are: number of trees = 150, maximum tree depth = 6, number of randomly sampled variables = 4, learning rate = 0.3, and L2 regularization = 1.0.

### 5.2. Variable Importance and Number of Clusters in Cluster Analysis

Table 2 shows the variable importance in the cluster analysis. Overall, band 3 is the most important variable in the color dimension and PanStd is the least important. The exception is the greenhouse, which is black or white, with different spectral channels having similar weights. In the shape dimension, area and length are the most important variables, especially for complex roof buildings.

Figure A2 shows in detail the distribution of buildings with different roof types in the color and shape dimensions, and how the number of clusters was determined by statistical indicators. The first column shows the standardized values of the multispectral bands and the standard deviation of the panchromatic band, as well as the number of buildings in different clusters in the color dimension. The second column shows the scatter plots and histograms of the data for different color variables and the values of the Calinski-Harabasz index for different numbers of clusters. The third column shows the standardized values for area, length, width, and ratio, as well as the number of buildings in different clusters in the shape dimension. The fourth column shows the scatter plots and histograms of the data for different shape variables and the values of the Calinski-Harabasz index. In the color dimension, the value of the Calinski-Harabasz index for pitched roof buildings decreases as the number of clusters increases, so the optimum number of clusters is the smallest number of clusters, two. This divided the pitched roof buildings into two types of high and low brightness, with a higher number of brighter buildings (R1C1). The situation is similar for greenhouses and complex roof buildings, but there are more low brightness buildings than high brightness buildings. In the color dimension, the value of the Calinski-Harabasz index for color steel buildings reaches its maximum when the number of clusters is three. Therefore, the color steel roof buildings were divided into three types in the color dimension, corresponding to different colors. Unlike the color steel roof buildings, the flat roof buildings are monochromatic in color but vary in brightness. In the shape dimension, the pitched roof buildings were divided into three types, distinguished mainly by area and length. Greenhouses were simply divided into small and large types. The Calinski-Harabasz index for the other three roof types increases with the number of clusters, suggesting that they have very diverse shapes. We set the number of their shape clusters to 4, resulting in one large area type, one small area type, and two medium area types with different aspect ratios.

## 6. Conclusions

In this study, we performed a detailed classification of building types in rural areas. The data used were from a single imaging process of GF-7 satellite. The experiment was conducted in the rural areas of Pingquan in northern China. A two-stage method combining supervised classification and unsupervised clustering was used. In the supervised classification stage, our study shows that the XGBoost model constructed using spectral information and several morphological indicators extracted from building footprint performs well in roof type classification, with overall *Recall*, *Precision*, and *F*1*-Score* of 0.88, 0.91, and 0.89, respectively. We proposed a template-based height correction method for pitched roof buildings. It combined the geometric features of the building footprint, the street view photos, and the building height information extracted from the GF-7 stereo image, and reduced the RMSE of pitched roof building heights from 2.28 to 1.20 m. In the unsupervised clustering stage, different clusters were distinguished in the color and shape feature spaces according to statistical indicators. The fine-grained building types were represented by codes that could be traced back to the building’s roof type, color, shape, and height features. Some irregularly shaped buildings were neglected in this study, which needs to be improved in the future. Given the wide variety of topographic and architectural features in different rural areas, the next step is to test the fine-grained building classification method in more areas, especially in southern China, where rural buildings tend to be denser and mixed with vegetation. In this scenario, the use of higher resolution images will be helpful to reduce the ambiguity of the classification. We will also try deep learning models, such as those based on transformer architecture and attention mechanisms. Buildings are the most fundamental element of disaster management. The fine-grained building classification product can be used in conjunction with other types of information to assess vulnerability and loss in rural areas. It provides a basis for relocation, the construction of protective facilities and early warning systems, the assessment of insurance needs, and helps to improve the decision-making process during emergency response.

## Figures and Tables

**Figure 1 sensors-25-00392-f001:**
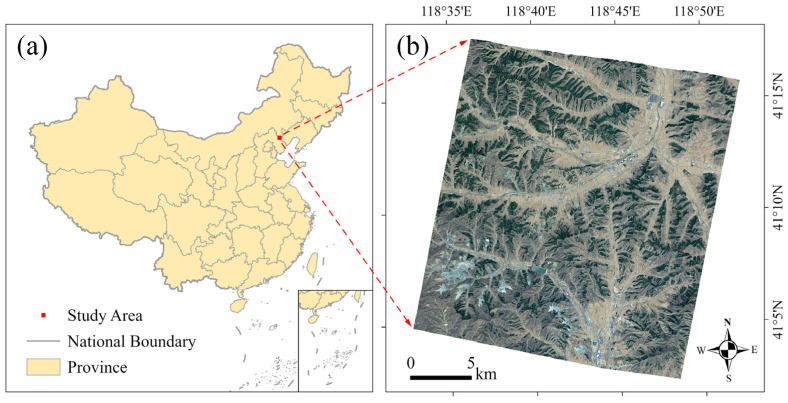
The study area: (**a**) location of the study area; (**b**) GF-7 MUX multispectral image of the study area.

**Figure 2 sensors-25-00392-f002:**
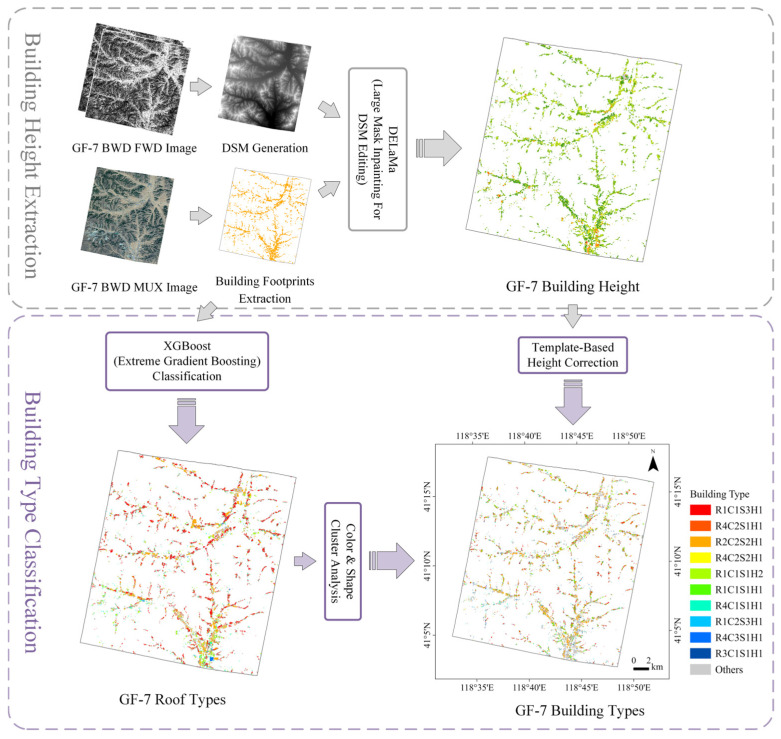
Workflow of the fine-grained building classification. The process of extracting building height from GF-7 data is also demonstrated.

**Figure 3 sensors-25-00392-f003:**
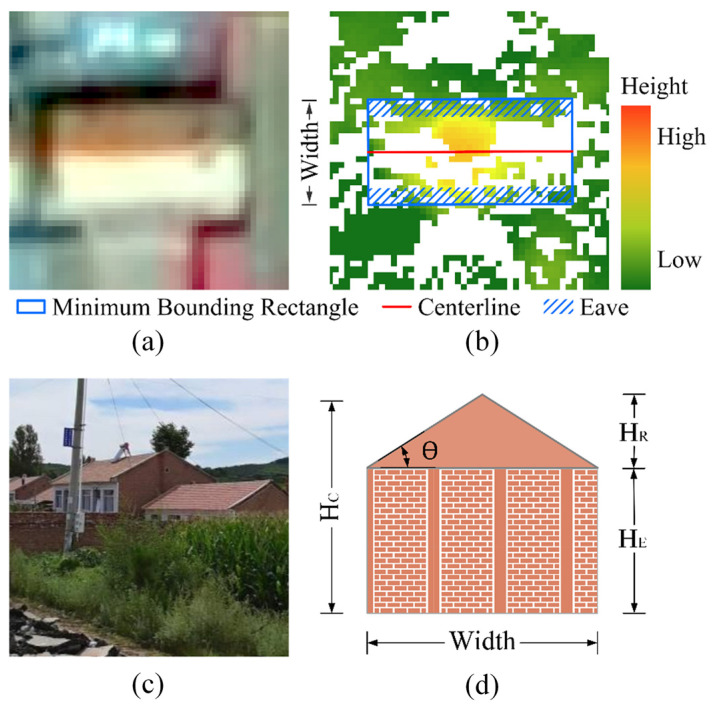
Template-based height correction for pitched roof buildings: (**a**) image of the pitched roof building; (**b**) minimum bounding rectangle and the eave zone; (**c**) street view photo of pitched roof buildings; and (**d**) template used for height correction.

**Figure 4 sensors-25-00392-f004:**
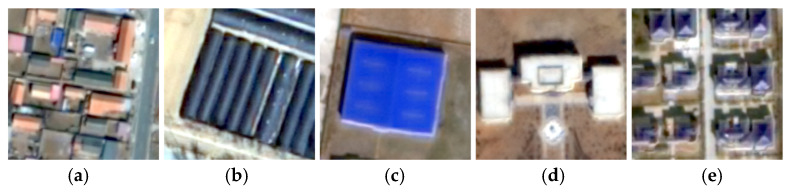
Samples of different roof types: (**a**) pitched; (**b**) greenhouse; (**c**) color steel; (**d**) flat; (**e**) complex.

**Figure 5 sensors-25-00392-f005:**
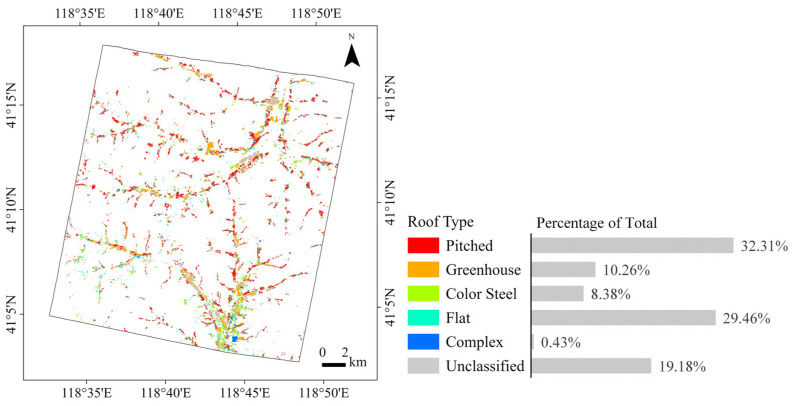
Roof types in the study area. Aggregated to 50 m ground sampling distance (GSD) for visualization.

**Figure 6 sensors-25-00392-f006:**
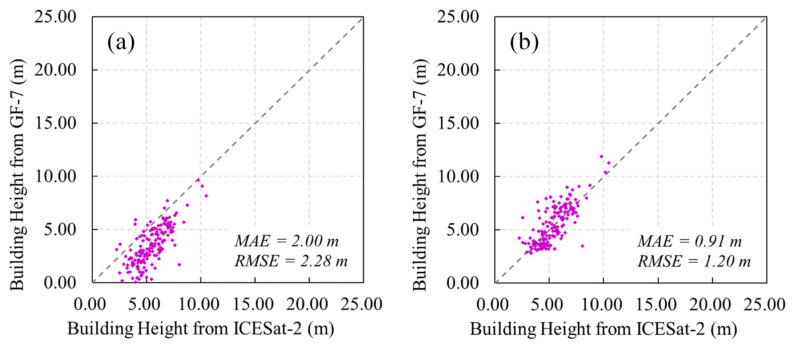
Height validation of pitched roof buildings: (**a**) before correction; (**b**) after correction.

**Figure 7 sensors-25-00392-f007:**
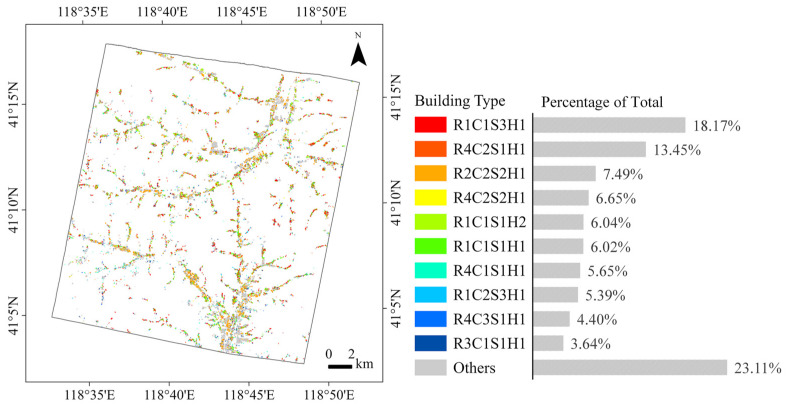
Fine-grained building types in the study area. Aggregated to 50 m GSD for visualization.

**Figure 8 sensors-25-00392-f008:**
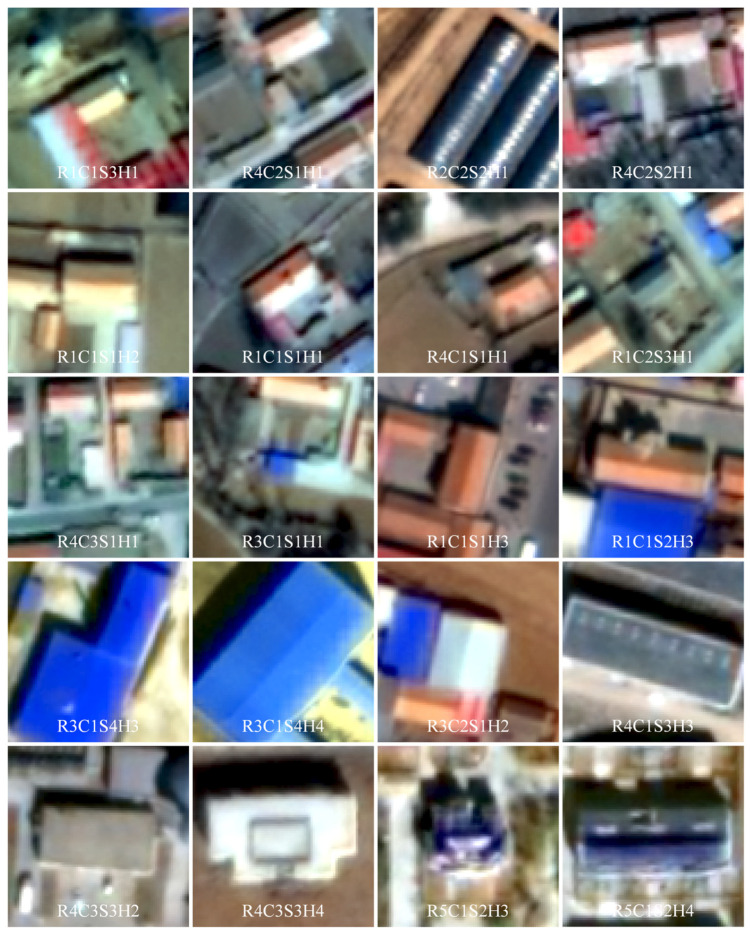
Samples of several representative building types.

**Figure 9 sensors-25-00392-f009:**
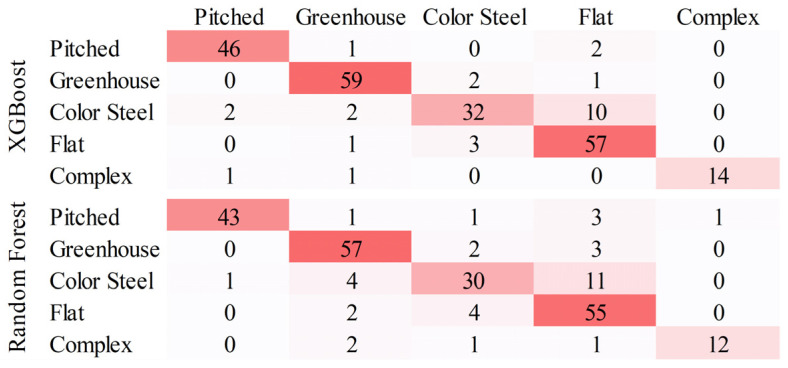
Confusion matrices of different supervised classification models.

**Table 1 sensors-25-00392-t001:** Evaluation indicators of different supervised classification models.

		Pitched	Greenhouse	Color Steel	Flat	Complex	Average
XGBoost	*Recall*	0.9388	0.9516	0.6957	0.9344	0.8750	0.8791
	*Precision*	0.9388	0.9219	0.8649	0.8143	1.0000	0.9080
	*F*1-*Score*	0.9388	0.9365	0.7711	0.8702	0.9333	0.8900
Random Forest	*Recall*	0.8776	0.9194	0.6522	0.9016	0.7500	0.8201
	*Precision*	0.9773	0.8636	0.7895	0.7534	0.9231	0.8614
	*F*1-*Score*	0.9247	0.8906	0.7143	0.8209	0.8276	0.8356

**Table 2 sensors-25-00392-t002:** Variable importance in cluster analysis.

Roof Type	Color					Shape			
	Band 1	Band 2	Band 3	Band 4	PanStd	Area	Length	Width	Ratio
Pitched	0.2500	0.4519	0.6198	0.6025	0.3206	0.6598	0.6786	0.5351	0.3687
Greenhouse	0.8503	0.8529	0.8327	0.7515	0.6127	0.5844	0.5723	0.1760	0.3683
Color Steel	0.5745	0.6019	0.7901	0.6334	0.2487	0.7033	0.6969	0.6735	0.5786
Flat	0.5451	0.6677	0.7185	0.6386	0.3519	0.7198	0.6930	0.4608	0.5493
Complex	0.6131	0.6527	0.7042	0.3101	0.0654	0.7932	0.8126	0.5090	0.6622

## Data Availability

Data available on request due to restrictions.

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
