# Peer review of "Fine-Grained Building Classification in Rural Areas Based on GF-7 Data"

_sensors, 2025, doi:10.3390/s25020392_

Round 1
Reviewer 1 Report
Comments and Suggestions for Authors
1. In the introduction section, the authors might present the study significance and knowledge gap with a logical line, not commend to say xxx authors did some kind of research in a specific year. These occassions occured several times, I suggest the authors correct these statements.
2. I would suggest the authors give very clear and specific objectives for this research and delete the last paragraph in section 1.
3. In section 3.2, it would be better to give some specific statements for RF and deep learning model, e.g., ResNet-34. Also, some modification or improvement or special consideration for these modeling methods are strongly proposed.
4. In the results section, is it possible to give defination for your rooftype, and also give examples how these rooftype differentiated to each other, in spectral, texture, and other aspects as well.
5. In figure 5, you may remove R2 in the subplots, for the validation, RMSE and bias or MAE are important proxies for evaluation model accuracy.
Comments on the Quality of English LanguageThe English language is relatively good, there still room to improve.
Reviewer 2 Report
Comments and Suggestions for Authors
The manuscript "Fine-Grained Building Classification in Rural Areas Based on 2 GF-7 Data " adopted high-resolution dataset into accurate building classification with machine learning approaches. The authors have presented a well-structured and comprehensive study that offers valuable insights into combining various remote sensing techniques.
After reviewing the paper, I find no major issues or concerns that would prevent its acceptance. But some questions should be addressed to make the paper robust.
(1) Although the accuracy of the XGBoost classification model is high, have the authors repeated the model using different random sets of training/testing datasets and investigated if the results are robust? Are there any possibilities of overfitting?
(2) When applying ML tools, it is crucial to provide several quality metrics such as ROC-AUC and PR-AUC and conduct many comparisons among the parameters to make a solid decision, especially in an unbalanced dataset, did the authors check these parameters?
(3) The topography in the study area changes a lot and there are few villages in this area. As the goal of the results in this study aims at supporting disaster management, what about the area with densely distributed buildings? Will there be any ambiguity to make classification if the topography and land surface changes?
Reviewer 3 Report
Comments and Suggestions for Authors
Comments
Article Fine-Grained Building Classification in Rural Areas Based on GF-7 Data submitted by authors: Mingbo Liu, Ping Wang, Peng Han, Longfei Liu to Sensors dedicated to the topical problem of obtaining detailed information on building type which is widely used in risk and loss assessment of natural disasters such as earthquakes, landslides and floods as well as in studies related to urbanization, energy consumption and population modeling.
The objective of this study is to assess the capabilities and limitations of building footprints, building heights, and multispectral in formation extracted from GF-7 satellite data for fine-grained building classification in rural areas. Authors also proposed a method for correcting the height bias of pitched roof buildings.
The diversity of topography and industry makes the study area a representative sample. The authors performed an extensive manual correction of the extracted building footprints through visual interpretation of the multispectral image.
A building height extraction method that combines photogrammetry and deep learning is proposed. The authors used a two-stage method combining supervised classification and unsupervised clustering for fine-grained classification of buildings.
The authors in this paper manually labeled 1171 samples of different roof types including pitched, greenhouse, colored steel, flat and complex roofs
It was found that pitched roofs and flat roofs are the most common roof types, accounting for more than 60% of the total, followed by greenhouses and colored steel roofs.
There are some questions and recommendations:
1.Which building classification is more optimal and disaster resistant in rural areas?
2.Is it possible to compare the data obtained by the authors with the data of other authors?
3.In conclusion - “and F-score of 0.8791, 394 0.9080, and 0.8900, respectively” should be replaced by “and F-score of 0.88, 0.91 and 0.89, respectively”.
4.The latitude of the study area is 41°2'27 “N~- 41°17'49 ‘N and your model indicates an average slope of 32.5°. What will be the result if the average slope will be 41°9’?
5.Since rural areas are mostly affected by natural disasters, what recommendations can be offered to the population?
Hope that the work will be continued by the authors, taking into account the association of the characteristics of different types of buildings with disaster risk indicators as indicated by the authors in the conclusion.

Reviewer 4 Report
Comments and Suggestions for Authors
I have carefully read the paper, and while the topic is quite interesting, the methods are relatively simple, though they include some small innovations. The paper has the potential to be published in the journal after some revisions.
In line 41, please add the year of the citation. Also, for Atwal et al., MemduhoÄŸlu et al., and others, please check the entire paper, as I couldn't find the year of the citations.
In the literature review section, lines 41 to 56, please add the key results of the reviewed papers, such as OA, and any other relevant findings.
I believe the introduction requires a thorough revision. The literature review is incomplete, as it is missing important information from previous studies, such as key findings, suggestions, and identified gaps. Additionally, the research gap is not clearly highlighted, and the significance of your research is not well explained. It would be helpful to discuss the advantages your study offers in fine-grained building classification, particularly in comparison to previously published papers.
In the section 'The 90th percentile value of the nDSM in the building footprint was then used as the building height,' why did you choose the 90th percentile? Why not the 95th or 100th percentile? How did you determine this threshold? Was there any validation process or testing involved in selecting this value?
The quality of Figure 2 is low and does not meet the standards required for publication.
Provide more information about the hyperparameter tuning for models such as XGBoost. Specifically, which parameters did you tune? Did you use cross-validation, or was another method employed to optimize the model?
It would be helpful if you could compare the differences between the different models statistically. For example, you could use the McNemar test to determine if there are significant differences between them. Conducting such tests, especially at the class level, would provide valuable insights into the limitations of the methods and the GF datasets for classification in each class.
